# Evaluation of innovation primacy in cross-regional central cities: Evidence from the Huaihai Economic Zone in China (2010–2020)

**Qin-Xia Liu** *

School of Economics and Management, Jiangsu Vocational Institute of Architectural Technology, Xuzhou, Jiangsu, China

* lqxia2004@163.com, 872969467@qq.com

## Abstract

With the intensification of global economic competition, innovation has become one of the core elements of vigorous development in various regions. Improving the innovation ability of cross-regional central cities is the main factor influencing whether a region can achieve economic and social development. In this paper, an innovation primacy index system is designed according to the links of the innovation value chain, and the innovation advantages and empirical effects are comprehensively analyzed by using the point-to-point method, entropy weight method, gravity model and two-way fixed effect model. Based on the data of 8 cities in the core area of the Huaihai Economic Zone from 2010 to 2020 in China, Xuzhou's innovation primacy ranks first in the core area of the Huaihai Economic Zone, which accords with its status as the central city of the region. Its index has been rising, and its development trend is also good. However, the overall innovation ability of the core area of the Huaihai Economic Zone is unbalanced; the overall economic foundation is not solid enough. In terms of knowledge innovation, R&D innovation and industrial innovation, the industrialization level, around the activation of Xuzhou city vitality, enhances the Xuzhou innovation radiation drive, leading to a decrease in the Huaihai Economic Zone. Finally, some corresponding suggestions on innovation primacy have been proposed for the city of the Huaihai Economic Zone.

## Introduction

Improving innovation capacity is a key driver of economic and social development. With the intensification of global economic competition, innovation has become one of the core elements for countries and regions to thrive [1–3]. City clusters are the main form of China's new urbanization development and are important spatial carriers of international competition and cooperation. As an economic core area spanning multiple provinces, Xuzhou is an economic core city at the junction of four provinces in China. There are eight core Huaihai economic cities in the Huaihai Economic Zone, namely, Xuzhou, Jining, Zaozhuang, Lianyungang, Suqian, Shangqiu, Suzhou and Huaibei. It aims to create a new engine for economic take-off across

**Data Availability Statement:** This text is appropriate if the data are owned by a third party and authors do not have permission to share the data. The authors confirm that others would be able to access these data in the same manner and

confirm that the authors did not have any special access privileges that others would not have. The sources of data acquisition are described as follows: 1)Knowledge innovation index data. The number of SCI, CPCI-S and SSCI papers was used as the evaluation index. The specific search methods used were as follows: the corresponding subdatabases in the Web of Science database were selected, and the country (CU=China), province/state (such as PS=Jiangsu), and city (such as CI=xuzhou) were set in order from the advanced search bar and year of paper publication. From these, the annual numbers of papers published in three indexes of each province, region and city in the Huaihai Economic Zone were obtained. 2)R&D innovation index data. The number of patent applications, the number of patent grants and the number of valid invention patents are selected for the evaluation system. The data come from the Patent Statistical Annual Report of the State Intellectual Property Office, local statistical yearbooks, science and technology statistical bulletins, etc. 3)Industrial innovation index data. The number of enterprises, the output value of high-tech industry and the sales revenue of new products are selected as the evaluation indicators. The data were obtained from the China Torch Statistical Yearbook, local statistical Yearbook and local science and technology statistical bulletin.

**Funding:** The General Project Subjects of Philosophy and Social Science Research in Universities of Jiangsu Province, Evaluation and influencing factors of green technology innovation of listed enterprises in Jiangsu Province under the background of digital transformation (Grant No. 2023SJYB1177).

**Competing interests:** The authors have declared that no competing interests exist.

provincial borders, improving the region's innovation capacity as the power source for the region to achieve great economic and social development. The enhancement of innovation capacity will help attract more investment, talent and resources and drive the rapid growth of the regional economy. In 2017, Xuzhou was planned and deployed as a separate important plate in the province's "1+3" functional area strategy, and opinions were issued to support the construction of the central city of Xuzhou in the Huaihai Economic Zone. The ability of the Huaihai Economic Zone to promote coordinated development at the national level is of highly strategic significance for overcoming administrative barriers, promoting the rise of depression, and supporting coordinated national innovation development. The Huaihai region is densely populated with small and medium-sized towns, and the linkages and cooperation between towns are extensive and close, especially Xuzhou, as the center of the regional town system is becoming increasingly prominent. In 2022, the total number of Xuzhou's seven major economic indicators will be at the forefront of the 19 cities in the Huaihai Economic Zone. The city's GDP will reach 845.784 billion yuan, ranking first in the region and accounting for 11.7% of the regional share; retail sales of consumer goods will reach 410.273 billion yuan, ranking first in the region and accounting for 12.7% of the regional share; the balance of RMB deposits in financial institutions will reach 105.5362 billion yuan, ranking first in the region and accounting for 10.3% of the regional share; and the pattern of regional development centered around Xuzhou will be accelerated. As the leading city of the Huaihai Economic Zone, Xuzhou's innovation primacy can reflect its influence and influence on the social development of the city. In-depth analysis of Xuzhou's innovation capacity performance can provide useful experience and inspiration for other cities and promote common development. Moreover, Xuzhou needs to enhance the first place of science and technology innovation from various aspects to promote the common development of the Huaihai Economic Zone. In this context, analyzing the current situation and problems related to the innovation primacy of the core cities in the Huaihai Economic Zone is highly important for exploring how Xuzhou plays a leading role in the Huaihai Economic Zone and how to enhance the innovation capacity of the central cities in the Huaihai Economic Zone. Therefore, it is highly important to strongly support the construction of Xuzhou as the central city of the Huaihai Economic Zone, to enhance the comprehensive function and innovation primacy of Xuzhou's central city, and to actively tilt toward Xuzhou in terms of coordinating the study of the major infrastructure and major productivity layout and the major carriers and platforms to further enhance the function of radiation and drive the regional central city of Xuzhou and to practically lead the collaborative and innovative development of the Huaihai City cluster.

Therefore, this paper selects the data of eight cities in the core area of the Huaihai economy from 2010–2020 and uses various methods to analyze and evaluate innovation primacy in stages. Specific evaluation methods: First, the point-to-point method is used to calculate and evaluate the innovation primacy. The main reason for adopting the point-to-point method is that the indicators used in this paper are designed according to the three stages of the innovation value chain, the first-level innovation cities at different stages are not controllable, and the development stages of each city in the country are different. There is no way to achieve complete uniformity of data statistics, and comparability is poor; however, the caliber of statistics is the same at each provincial scale. Therefore, this paper first adopts the point-across method to calculate the data of the three-level indicators; that is, the proportion of the city indicator data in the province is used to measure the concentration of the target city's innovative activities in the province [4]. Second, the entropy weighting method is used to calculate the proportion of each indicator; finally, the evaluation score of each city's innovation popularity is calculated, and GIS technology and social network analysis are used to measure the core city's innovation capacity. Third, a two-way fixed effect model is constructed to empirically analyze the

implementation performance of Xuzhou's innovation-leading effect to answer the question of whether Xuzhou plays an innovation-leading role in its neighboring cities.

The significance of the research is as follows: First, the analysis and evaluation of innovation propagation through various methods can help to deepen the understanding of the characteristics and differences of innovation activities in various cities and help to formulate more targeted policies and strategies to promote the overall improvement of innovation ability. Second, building a two-way fixed effect model and conducting an empirical analysis of the innovation-leading effect of Xuzhou City will help us determine whether Xuzhou City has played a role in innovation-leading. Answering this question will help assess the real impact of leading cities across economic zones and provide strong support for future policymaking. Finally, the innovation primacy of the core cities in the Huaihai Economic Zone has also been researched in depth. In terms of knowledge innovation, R&D innovation, industrialization level and industrial innovation, Xuzhou city vitality should be activated, and the Xuzhou innovation radiation drive should be enhanced to increase the "depression rise" in the Huaihai Economic Zone. Therefore, the results of this study have important theoretical and practical value.

## Literature review

### Innovation primacy

According to Jefferson [5], primacy refers to the ratio of a country's (or region's) population in the first city to that in the second city. Primacy can be calculated using the relatively simple and practical two-city index method, as well as the "four-city index", "eleven city index" and point-to-point method. Several studies have shown that the results of these two methods are strongly correlated. Later, Marshall [6] proposed a specific quantitative criterion for the primacy index. He believed that a primacy index of 2 is a more reasonable distribution, a distribution between 2 and 4 is a medium primacy distribution, and a distribution greater than 4 is a high primacy distribution. Chinese scholars have conducted a wealth of research on primacy [7, 8]. From the initial demographic field to the economic, industrial, scientific technological, and public service fields, cities' primacy and influencing factors have been measured in many dimensions and at many levels. In recent years, scholars such as Zeng [9] and Li and Zhou [10] have gradually focused on the primacy of innovation. Guan and Sun [11] analyzed the current situation and existing problems of Nanjing's innovation primacy and proposed that Nanjing focus on improving its basic research capacity, strengthening the function of market resource allocation, improving its financial support capacity, and building a regional open innovation system. International researchers have focused on competition and cooperation between cities, and studies have shown that cities with high primacy usually have advantages in attracting talent, capital, and innovation resources [12]. As a result, studies have emphasized the close relationship between urban primacy and urban competitiveness. Several studies [13, 14] have noted that science, technology, and innovation are important engines for driving economic growth and social development in cities. Silicon Valley, as a world center of science, technology, and innovation, is an example of a city that has attracted many high-tech enterprises and innovative start-ups, which has led to the prosperity of the region. This model of an innovation-focused city is widely used internationally.

### Innovation value chain

Jefferson [15] proposed a more systematic theory of the innovation value chain. They believe that the innovation value chain consists of three links—i.e., idea generation, idea transformation and idea diffusion—and emphasize that these three links are interdependent and that any weak link directly affects the whole innovation process and the overall innovation effect. This

theory has been widely used in innovation research [16, 17] and by Li et al. [18] based on the three links of knowledge innovation, research innovation and product innovation in the innovation value chain. Using the DEA model to evaluate and analyze the innovation efficiency of each stage, it was found that the inputs and feedbacks of the innovation elements have a more obvious impact on the whole innovation process. Doran and O'Leary [19] established an evaluation index system that included three aspects—knowledge innovation, research and development innovation and product innovation—based on the results related to the innovation value chain of previous researchers and evaluated and compared the innovation primacy of Nanjing with that of other central cities by using principal component analysis and a primacy matrix. Some studies also emphasize the important role of city governments; city policies and strategies can promote innovation primacy in cities by encouraging innovation, providing entrepreneurial support, and improving infrastructure [3, 20]. For example, Stockholm, Sweden, has undertaken several policy initiatives to establish the city as a global center for eco-technology and innovation, with notable success. The establishment of global city networks [21] has made it easier for cities to innovate and collaborate. Researchers [22] have studied the patterns of cooperation between global cities, emphasizing the importance of the formation of cross-city innovation networks for city primacy.

In general, there have been an increasing number of studies on city primacy in recent years, and the theories and methods used have become more comprehensive and richer. Compared with the existing studies, the possible marginal contributions of this paper are threefold: (i) There are relatively rich studies on innovation primacy and few studies on innovation primacy from the perspective of the innovation value chain. This paper evaluates and analyses the innovation capacity of core cities in the Huaihai Economic Zone from the perspective of the innovation value chain by combining various methods, to provide a scientific basis for the improvement of Xuzhou city's innovation primacy; (ii) the literature only focuses on the leading effect of the central city on the noninnovation variables of the surrounding cities, but less research on the innovation-leading effect of the central city on the innovation capacity of the surrounding cities, this study is a useful supplement to the literature; (iii) realistically, Xuzhou's economic development is indeed superior to that of its neighboring regions. Controlling for industry and year as well as other factors, can the innovation-leading effect of Xuzhou, as a central city on its neighboring cities, be tested empirically and quantitatively, how does it perform under the analysis of robustness and heterogeneity, and what mechanisms can be used to influence the innovation of its neighboring cities? These analyses can provide empirical insights for governments to develop differentiated innovation programs.

## Methodology

### Regression model setting

To further analyze whether Xuzhou plays a leading role and determine the difference in the impact of Xuzhou as a central city on cities of different sizes, the heterogeneity of Xuzhou as a central city in innovation leadership is analyzed. The measurement model is constructed as follows:

$$\text{Innovcap}_{it} = \beta_0 + \beta_1 \text{ Innovled}_{it} + \varphi \text{ Control}_{it} + \mu_i + \nu_t + \varepsilon_{it} \tag{1}$$

In particular, $\text{Innovcap}_{it}$ denotes the innovation capacity of city i at time t, which is the explanatory variable of the model; $\text{Innovled}_{it}$ denotes the innovation-leading effect of the central city on other cities at time t, which is the core explanatory variable of the model; $\text{Control}_{it}$ is the control variable; $\mu_i$ denotes the city fixed effect; $\nu_t$ denotes the time fixed effect; and $\varepsilon_{it}$ denotes the random error term.

## Setting the variables

**Explained variables.** The innovation primacy of the surrounding city (Innovcap): This is a measure of urban innovation ability calculated by the entropy weight method. Technological innovation is a continuous driving force for urban development. It can not only promote the rapid economic development of the city itself but also promote the rapid economic development of the region. According to innovation value chain theory, the innovation process mainly includes knowledge innovation, R&D innovation, and industrial innovation [15, 23]. Knowledge innovation is the most basic support and guarantee of regional innovation. R&D innovation is a bridge connecting knowledge innovation and industrial innovation. On the one hand, it verifies whether knowledge innovation is practical. On the other hand, it lays a solid technical foundation for industrial innovation. The primacy of urban innovation reflects the level of urban scientific and technological innovation, both reflecting the city's 'soft power' and enhancing the city's 'hard power'. Therefore, based on the results of the previous innovation first-degree evaluation index system and considering the availability of data, this paper designs the first-degree index system of scientific and technological innovation according to the three links of the innovation value chain [19, 24] (as Table 1). The concept of entropy was introduced in 1864 and was originally used to describe the state of a system. Shannon, the author of information theory, introduced the concept of information entropy and suggested that entropy can measure the uncertainty of information and can also be applied to other fields. As a result, entropy started to become a measure of uncertainty, and the entropy weight method is one of the relevant applications. The entropy weighting method determines the weight of each indicator by measuring the information content of each indicator in the system. Generally, there is a complementary relationship between the amount of information and entropy; that is, the higher the entropy is, the lower the amount of information.

## Core explanatory variables

Xuzhou's Innovation primacy (Innov). According to the related literature, the gravity model is widely used to measure city innovation-related effects. The gravitational model is used to measure the influence of Xuzhou city on innovation in neighboring cities, as shown in the following equation.

$$Innovled_{it} = \frac{KC_i C_j}{D_{ij}^2} \tag{2}$$

**Table 1. The evaluation index of innovation primacy in cross-regional center cities.**

| First level index | Second level index | Three level index | Method of calculation |
|---|---|---|---|
| Innovation primacy | Knowledge innovation primacy | Number of SCI papers | The proportion of SCI papers in the whole province |
| | | Number of SSCI papers | The proportion of SSCI papers in the whole province |
| | | Number of CPCIS papers | CPCIS total number of papers the province share |
| | R&D innovation primacy | Number of valid invention patents | The proportion of effective invention patents in the whole province |
| | | Number of patents granted | The province wide proportion of patent licensing |
| | | Number of patent applications | The province wide proportion of patent applications |
| | Industrial innovation primacy | Hightech zone torch plan statistics number of enterprises | Hightech Zone Torch Plan statistics the number of enterprises in the province |
| | | Hightech industry output value (billion yuan) | The high-tech industry output value of the province as a percentage |
| | | New product sales revenue (billion yuan) | Sales of new products accounted for the proportion of the province |

where *Innovled*$_{it}$ is the innovation-leading effect of central city i on peripheral city j, K is the gravity constant, $C_i$ is the innovation capacity of the central city, $C_j$ is the innovation capacity of the peripheral city, and $D_{ij}^2$ is the square of the spatial straight line distance between the central city and the peripheral city. Drawing on the relevant literature, to facilitate data processing, K = 1000.

## Control variables

By listing several control variables, other factors that may affect the innovation capability of a city can be controlled to ensure the accuracy of the analysis. The following is an explanation of each control variable and an explanation of the reasonableness of the selection: (i) Level of economic development (rpgdp) [25, 26]: The per capita gross national product (GNP) is an important index for measuring the economic prosperity of a region or city. It reflects the average economic value created by each resident and is a representative indicator of the overall economic level of a city or region. Economic status is often correlated with innovation ability. A region's economic prosperity may mean that more resources are available to support innovative activities, such as investment in research and development and talent attraction. (ii) Degree of opening up (import and export quota) [5, 27]: Openness refers to the extent to which a city or region is connected to international markets, usually measured by imports and exports. Highly open cities may have better access to international innovation resources and market opportunities. International trade and cooperation can bring new opportunities for innovation and promote technology dissemination and knowledge exchange. (iii) An increase in imports and exports may be related to a greater level of innovative activity development [19, 28] (balance of deposits and loans at the end of financial institutions): The level of financial development reflects the size and health of a region's financial system. Adequate financial resources and a developed financial market can support the innovative financing needs of enterprises. The financial system plays a key role in fostering innovation. It can provide access to finance, risk management and investment, helping to drive innovation. (iv) Technical development level (tech) [29–31]: The level of technological development usually refers to the level of scientific and technological research and development and innovation output of a city or region. Cities with high levels of science and technology may be better equipped to conduct cutting-edge technological research and innovation. Technological development level is directly related to innovation ability. Investment and progress in the field of science and technology can greatly affect a city's innovation performance.

Overall, these control variables are chosen because they represent key aspects of a city's economy, openness, finance, and technology, which are closely related to its ability to innovate. However, in specific models, the effects of these variables on innovation capacity may vary by region and require further study and interpretation.

## Data sources

The data sources for the explained variables are derived from the Web of Science database, the Patent Statistical Yearbook of the State Intellectual Property Office, the Regional Statistical Yearbook, the China Torch Statistical Yearbook and the Regional Statistical Bulletin on Science and Technology. Specifically, for the number of SCI, CPCIS and SSCI papers, the corresponding subdatabases were selected from the Web of Science database, and the country (CU = China), province/state (such as PS = Jiangsu), city (such as CI = Xuzhou) and paper publication year were assigned to the advanced retrieval column to obtain the annual number of papers published in the three types of indexes of provinces and cities.

The remaining data are from the 2010–2020 China Urban Statistical Yearbook, China Statistical Yearbook, Shandong Statistical Yearbook, Jiangsu Statistical Yearbook, Anhui Statistical Yearbook, China Science and Technology Statistical Yearbook, and Henan Statistical Yearbook, as well as statistical yearbooks and statistical bulletins of eight prefectural-level cities. In addition, to reduce the effect of heteroscedasticity, the data of some variables are logarithmic.

## Empirical analyses

### Evaluation and analysis of innovation primacy

This paper adopts the entropy method to measure the innovation capacity of core cities in the Huaihai Economic Zone from 2010 to 2020. Before the measurements, the secondary indicators were standardized to unify the indicator scale (as shown in Table 2). On this basis, the corresponding weights are assigned to the secondary indicators, and the individual scores of the secondary indicators are calculated to arrive at the comprehensive score of each city's innovation capacity.

Regarding the average score over the past ten years (as shown in Fig 1 and Table 3), Xuzhou's innovation ability ranks first, with 45.42 points. This is followed by the other cities: Jining, Suzhou, Huaibei, Zaozhuang, Shangqiu, Lianyungang and Suqian. Jining is second, with a score of 35.32 points. Jining's innovation primacy is relatively balanced, with a high score for R&D innovation and industry innovation, but knowledge innovation is slightly lower, which affects the overall innovation score. The third is Suzhou, which scored 20.22 points, indicating high knowledge innovation, low R&D innovation and low industrial innovation. The innovation primacy scores of these three cities are greater than 20 points, ranking high in the entire Huaihai Economic Zone, and these three cities are also important economic pillars of the Huaihai Economic Zone. However, there is a large gap between the innovation primacy of the top three cities, with a difference of 10 points. For other cities, the total innovation ability score is less than 20 points; thus, innovation ability needs to be continuously improved. From the perspective of subpriorities, most cities have the lowest knowledge innovation primacy, followed by R&D innovation primacy. Compared with other relatively large economic zones in China, the Huaihai Economic Zone does not have strong geographical advantages, and its industrial support and business environment are relatively weak. Improving the regional innovation environment is a special development route that has a great effect on revitalizing interprovincial regional economic resources.

The innovation primacy of average innovation or knowledge, R&D, industry innovation, and Xuzhou's score is at the forefront, indicating that Xuzhou's innovation primacy is the best

**Table 2. The innovation primacy index and its weight.**

| First level index | Second level index | Second level weight | Third level index | Third level weight |
|---|---|---|---|---|
| Innovation primacy | Knowledge innovation primacy | 48.84% | Number of SCI papers | 11.51% |
| | | | Number of CPCIS papers | 13.40% |
| | | | Number of SSCI papers | 23.93% |
| | R&D innovation primacy | 25.51% | Number of patent applications | 7.04% |
| | | | Number of patents granted | 7.44% |
| | | | Number of valid invention patents | 11.02% |
| | Industrial innovation primacy | 25.65% | New product sales revenue (billion yuan) | 7.04% |
| | | | Hightech zone torch plan statistics number of enterprises | 7.59% |
| | | | Hightech industry output value (billion yuan) | 11.02% |

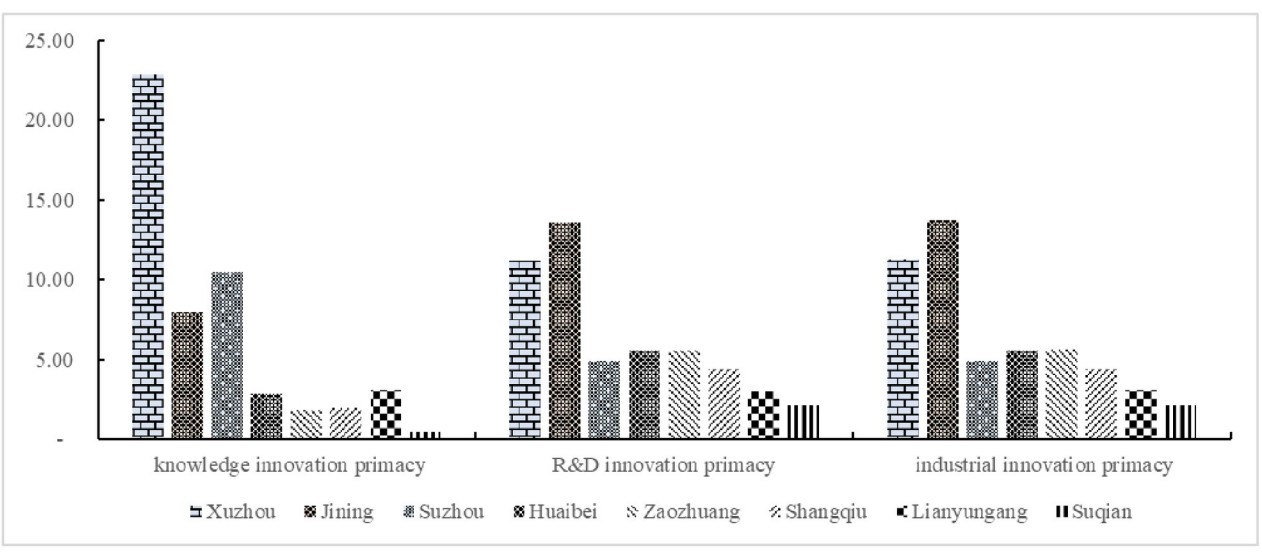

**Fig 1. The innovation primacy of core cities in the Huaihai Economic Zone.**

among the cities in the Huaihai Economic Zone, in line with its status as the core city of the Huaihai Economic Zone. Xuzhou has a strong radiation ability and leading ability to reach the Huaihai Economic Zone [9]. According to the subitem primacy score, Xuzhou's knowledge innovation primacy score is the highest at 22.87 points, and the industrial innovation primacy and R&D innovation primacy scores are 11.19 points and 11.27 points, respectively. This shows that the innovation primacy of Xuzhou is reflected mainly in knowledge innovation. Although the scores for industrial innovation and R&D innovation are relatively high, there is still a large gap compared with the scores for knowledge innovation primacy. Xuzhou needs to further balance the relationship between knowledge innovation and R&D innovation and enhance the ability of knowledge innovation to transform into market-oriented results. In general, with the first ranking of urban innovation primacy, Xuzhou must play a leading role in the Huaihai economic region, give full play to its advantages in the agglomeration of scientific and technological resources and the scientific and technological service system, and lead science and technological innovation and development in the Huaihai region as a whole.

Fig 2 shows that between 2010 and 2020, the innovation popularity of the core cities in the Huaihai Economic Zone generally exhibited a fluctuating upward trend. Among these cities, the first degree of innovation in Xuzhou shows a trend of first declining and then rising. The

**Table 3. Innovation primacy of core cities in the Huaihai Economic Zone.**

| City | Innovation primacy | Knowledge innovation primacy | R&D innovation primacy | Industrial innovation primacy |
|---|---|---|---|---|
| Xuzhou | 45.33 | 22.87 | 11.19 | 11.27 |
| Jining | 35.32 | 7.97 | 13.62 | 13.73 |
| Suzhou | 20.22 | 10.45 | 4.89 | 4.88 |
| Huaibei | 13.88 | 2.84 | 5.52 | 5.52 |
| Zaozhuang | 12.96 | 1.81 | 5.55 | 5.60 |
| Shangqiu | 10.83 | 2.01 | 4.42 | 4.41 |
| Lianyungang | 9.13 | 3.07 | 3.01 | 3.05 |
| Suqian | 4.64 | 0.43 | 2.09 | 2.12 |

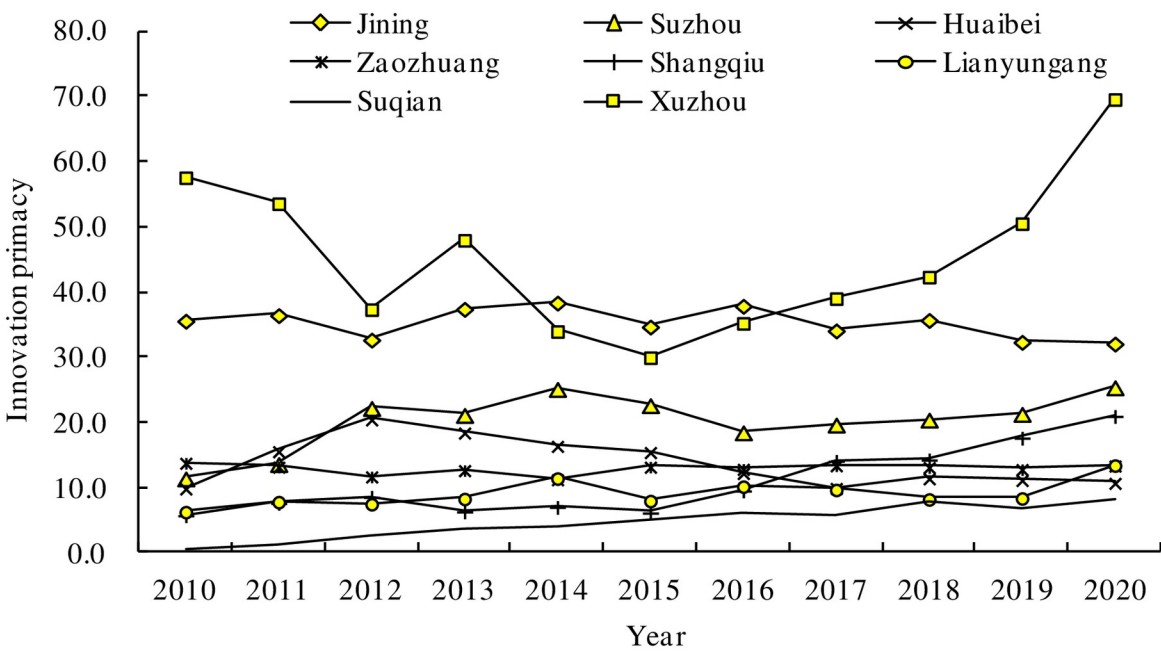

**Fig 2. Trend of innovation primacy in the core city of the Huaihai Economic Zone (2010–2020).**

turning point occurred in 2015, and it has been rising rapidly since 2015, reaching a maximum value of nearly 10 years by 2020, indicating that Xuzhou has focused on scientific and technological innovation in recent years. The implementation of the strategy of building a regional center has achieved remarkable results and has provided good momentum for development. Jining is relatively stable overall, showing a slight downward trend after 2016, but the level of innovation primacy has been at the forefront among the cities except Xuzhou in the Huaihai economic region over the past ten years. Suzhou was relatively backward before 2011, but with a rapid increase after 2011, it finally ranked third. There is a large gap between other cities and Xuzhou and Jining. Zaozhuang, Lianyungang and Shangqiu's innovation level is in the middle, although it fluctuates with a similar overall upward trajectory. Huaibei's innovation level is lower than before, and it began to rise after 2010 but with a downward trend after it reached its peak after 2012. Finally, it lags behind other cities in 2020, ranking only in front of Suqian. Suqian has been on the rise in the last ten years, showing a slight increase, but its overall innovation primacy is relatively low, ranking last. In general, the innovation ability of the core cities in the Huaihai region has been continuously improving over the past ten years, but the innovation ability gap between cities is large, and development has become unbalanced.

## Innovation leading effect of Xuzhou City

GIS technology is used to draw the innovation-leading effect maps of A.2010, B.2015 and C.2020, in which Xuzhou city is the central city and is set to 0. The larger the remaining values are, the stronger the innovation leadership is. Meanwhile, social network analysis(a.2010, b.2015,c.2020) is used to draw the innovation capability network map of the same year for comparison. The thicker the line, the stronger the innovation-leading effect of Xuzhou on the city.

As shown in Fig 3, Xuzhou is the innovation leader in the whole region; its innovation capability was already very high (88.39) in 2010, and its innovation leadership also occupied a

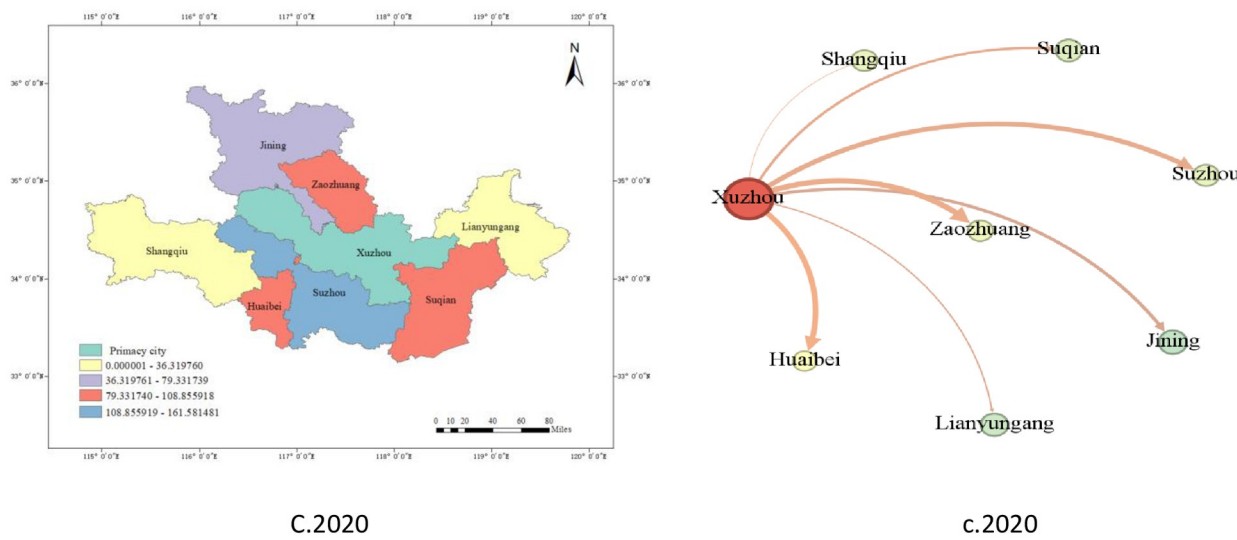

C.2020                                           c.2020

**Fig 3. Network distribution of spatiotemporal evolution value and innovation primacy of innovation leadership in Xuzhou City.**

dominant position. Despite a slight decline in innovation capacity in 2015 and 2020, it still maintains its leading position.

Time dimension: In 2010, Xuzhou City showed an excellent leading position in innovation ability (88.39). Due to the city's strong investment in technology, education and research and development over the past few years, it has attracted many innovative talent and enterprises. This moment marked Xuzhou's emergence in the field of innovation, becoming a leader in innovation in the region. In 2015, although Xuzhou's innovation capacity decreased slightly (67.33), it still held its leading position. Perhaps influenced by increased competition in the innovation sector, other cities are also actively developing innovation capabilities. At the same time, Xuzhou may face some challenges, such as competition for innovative resources and changes in market dynamics. This period reflects competition and cooperation in the field of innovation. In 2020, the innovation capacity of Xuzhou City increased again (78.32), consolidating its leadership position in the regional innovation field. This is mainly due to the continued increase in investment in technology, research and development and education by municipal governments and enterprises, which has attracted more innovative projects and enterprises. The 2020 data show Xuzhou's continued investment and results in the field of innovation.

Spatial dimension: This dimension reflects the comparison of innovation ability and innovation-leading effects among different cities in the Huaihai Economic Zone. As a central city in the region, Xuzhou has always been a leader in the field of innovation. Other cities, such as Zaozhuang, Jining, Huaibei, Suzhou and Lianyungang, have shown varying degrees of innovation strength in different years, and some cities have gradually increased their influence in the field of innovation. This comparison shows the dynamic development of the cities in the Huaihai Economic Zone in the field of innovation, as well as the gradual strengthening of the innovation cooperation network within the region. Specifically, with the above data, we can expand the analysis of the change trend of each city's innovation capacity and innovation leadership, as well as the possible causes and effects.

From Fig 4, innovation leads to the innovation of surrounding cities from 2010 to 2020. Every city has a specific number every year, which represents the number of a measure of the

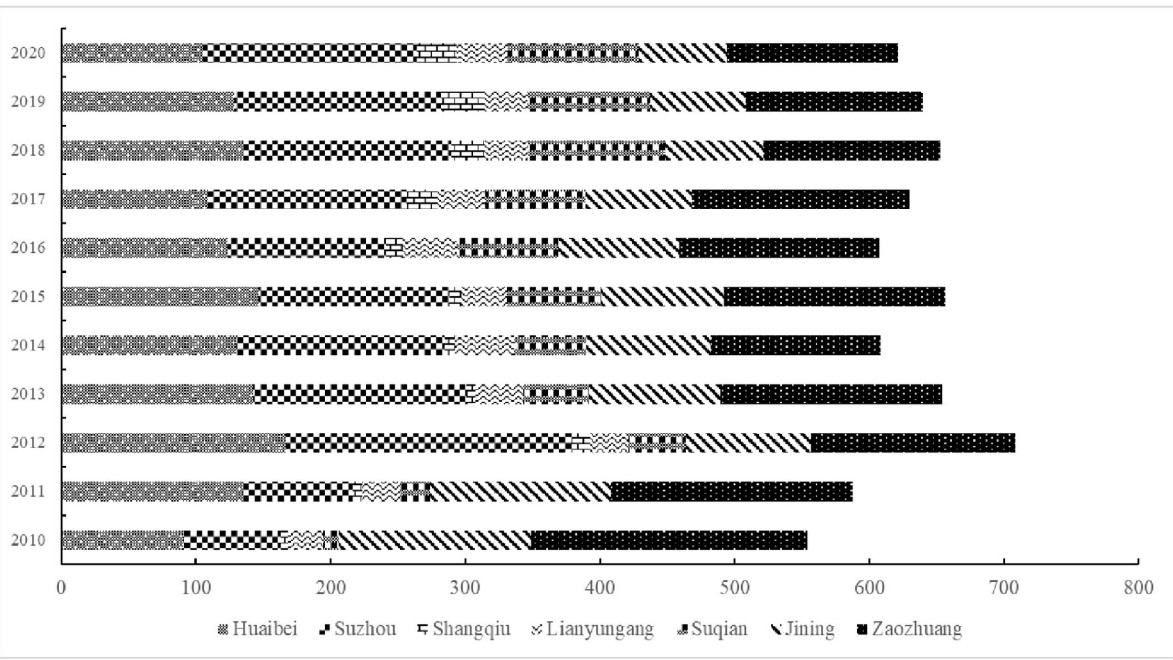

**Fig 4. Core cities in the Huaihai Economic Zone achieving to changes in trends through innovation.**

city in the year. Each part of the body represents a city, and the different colors of the body represent different cities. By observing the map, we can compare the numerical values of different cities in different years, and we can see the changing trends of each city in different years. The innovation ability and innovation of the surrounding cities can be found by comparing Fig 3 and the actual requirements.

1. Huaibei: 2010, at the middle level. However, by 2015, innovation and leadership had improved. This shows that the North's progress in innovation may involve attracting innovative resources through investment in technology and education. In 2020, however, innovation leads may be affected by the impact of competitive pressure and resource constraints.

2. Suzhou: In 2010, innovation reached moderate levels. By 2015, the innovation ability of Suzhou County had grown more. This may be because the state has increased its investment in innovative areas, attracting innovative projects and talent. By 2020, innovation capacity further improved to 9.60, and innovation led to 158.59. Shangqiu: In 2010, the annual watch was low, innovation ability was 1.32, and innovation lead was only 5.08. However, by 2015, innovation capacity improved to 2.93, and innovation lead increased slightly to 8.58. This shows that the Shangu has experienced certain growth in innovation, which may be achieved by improving education and R&D resources. By 2020, innovation capacity further increased to 8.87, and innovation lead reached 30.22.

3. Lianyungang: Lianyungang had a relatively high innovation lead in 2010 (25,73), although its innovation ability was lower (9.79). By 2015, innovation capacity had improved, and the innovation lead had increased to 33.69, indicating that Lianyungan had developed in the field of innovation. By 2020, innovation capacity further improved to 16.08, and innovation lead was 37.45. These findings indicate that the city has made progress in attracting innovative resources and projects.

In addition, Xuzhou's innovation-leading effect on Jining has declined over the past decade. It declined from 142.85 in 2010 to 64.88 in 2020, and Xuzhou's innovation-leading effect on Zaozhuang declined from 205.4041 in 2010 to 127.4288 in 2020, showing an overall decreasing trend. However, the innovation-leading effect of Xuzhou on Shangqiu and Suqian showed an overall increasing trend between 2010 and 2020. The different innovation-leading effects of cities reveal the differences in science and technology innovation, industrial development and economic growth among different cities. This difference stems from the differences in policy support, scientific and technological innovation capability, industrial structure, and talent introduction in different cities. Therefore, as a next step, the Xuzhou government, enterprises, and R&D institutions must conduct more in-depth studies and analyses, field research, and step-by-step actions to better cope with these differences and formulate corresponding policies and measures to promote the innovation-leading effect and to provide more targeted support and guidance for the promotion of scientific and technological innovation and economic development.

## Results analysis

### Benchmark regression results

Column (1) in Table 4 reports the regression results for Xuzhou's innovation-leading effect on surrounding cities. The estimated coefficients of the core explanatory variables are significantly positive at the 1% level, indicating that Xuzhou's innovation leadership has a positive effect on the improvement of the innovation capability of surrounding cities; that is, Xuzhou can promote the improvement of the innovation capability of surrounding cities through the innovation leadership effect. In terms of control variables, the economic development level of

**Table 4. Benchmark results for innovation-led core cities in the Huaihai Economic Zone.**

| Variables | (1) | (2) | (3) | (4) |
|---|---|---|---|---|
| Involved | 0.0839*** | 0.0839*** | 0.1010*** | 0.0542*** |
| | (0.0090) | (0.0067) | (0.0214) | (0.0169) |
| rpgdp | 7.6021 | 7.6021 | 5.6389 | 5.4585 |
| | (5.3436) | (4.8242) | (11.9024) | (5.2985) |
| ie | -21.8246** | -21.8246** | -24.5145*** | -18.8141** |
| | (8.7152) | (8.9252) | (8.1975) | (8.5980) |
| fin | -17.6171** | -17.6171** | -18.9922** | -14.6010* |
| | (7.4153) | (7.1753) | (8.6220) | (7.3567) |
| tech | 13.7881*** | 13.7881*** | 7.1979** | 13.3109*** |
| | (3.2238) | (3.3580) | (2.7824) | (3.1423) |
| City scale | | | | 0.0409** |
| | | | | (0.0199) |
| Constant | 4.8847* | 2.5557*** | 7.3159 | 5.4172* |
| | (2.7784) | (0.6830) | (7.4986) | (2.7132) |
| i.city | control | control | control | control |
| i.year | control | control | control | control |
| N | 77 | 77 | 42 | 77 |
| R² | 0.9743 | 0.9743 | 0.9825 | 0.9762 |

Note:

***, **, and * in the table indicate that the significance levels of the measurement results are 1%, 5% and 10%, respectively, and the values in brackets are robust standard errors.

the surrounding cities has no significant impact on their innovation ability; the import and export value and financial level negatively affect the innovation ability of the surrounding cities; and the scientific and technological development level significantly promotes the improvement of the innovation ability of the surrounding cities. The relationship between involved and other variables: The coefficient of involved is 0.0839***, which is statistically significant, indicating a positive relationship between involved and other variables. This means that there is a correlation between the innovation-leading effect of Xuzhou City and other variables and that an increase in other variables may promote the enhancement of the innovation-leading effect. The coefficient of the control variable, rpgdp (per capita gross national product), is not significant, indicating that in this model, per capita GNP has no significant influence on innovation-leading effects. Both io (degree of openness to the outside world) and fin (level of financial development) are significantly negatively correlated, indicating that degree of openness to the outside world and level of financial development are negatively correlated with the innovation-leading effect. Moreover, opening to the outside world and excessive financial development will have a negative impact on innovation-leading effects. The tech (technology development level) coefficient is positive and significant, indicating that the technology development level is positively correlated with the innovation-leading effect; that is, improving the technology development level may promote the enhancement of the innovation-leading effect. The coefficient of the city scale is positive and significant, indicating that city size is positively correlated with innovation-leading effects and that larger cities may be more conducive to the formation of innovation-leading effects.

To ensure the reliability of the results, a robustness test is carried out in this paper. ① Replace the measurement model. Due to the problems of simultaneous correlation and inter-group heteroscedasticity, the regression results may be biased when the bidirectional fixed effects model is used for analysis. In this paper, the panel-corrected standard error regression model is used to re-estimate the data, and the results are shown in column (2) of Table 4. The results show that the estimated coefficient of Xuzhou's innovation leadership effect is significantly positive at the 1% level, indicating that Xuzhou's innovation leadership has a positive effect on improving the innovation ability of surrounding cities, which verifies the robustness of the baseline regression results. ② Adjust the sample period. The sample duration of panel data will affect the baseline regression result, and the regression result may change when the sample duration is shortened or extended. In this paper, the sample period is adjusted from 2015 to 2020. According to the above analysis, 2015 was the turning point in the fluctuation trend of innovation primacy in the core cities in the Huaihai Economic Zone. Therefore, we focus on the innovation-leading effect of Xuzhou on the innovation ability of neighboring cities after 2015, as shown in column (3) of Table 4. The results show that the estimated coefficient of the Xuzhou innovation-leading effect is still significantly positive, which is consistent with the baseline regression result, indicating that the baseline regression result is robust.

To analyze whether the innovation-leading effect of Xuzhou City on neighboring cities is heterogeneous due to city size, this paper assigns a value of 1 to the city virtual variables of megacities and megacities in the Huaihai Economic Zone according to the Notice on Adjusting City Size Classification Standards issued by The State Council in 2014. Cities with populations greater than 5 million at the end of the year are assigned a value of 1; otherwise, 0. On this basis, the interaction term between the virtual variable of city size and the innovation-leading effect of Xuzhou City is included in the benchmark regression model to explore whether the innovation-leading effect of Xuzhou City is heterogeneous due to differences in city size. The results of the heterogeneity regression for city size are reported in column (4) of Table 4. The results show that the interaction term coefficient is significantly positive at the 1% level, indicating that Xuzhou's innovation leadership has a more significant effect on the innovation

ability of larger cities. Larger cities have strong scientific research strength, large R&D investments and excellent innovation environments, which are conducive to gathering more high-quality innovation factors and resources, better carrying out innovation cooperation with Xuzhou, and promoting overall innovation ability. In contrast, smaller cities are at a disadvantage in terms of scientific research strength, R&D investment, innovation environment, etc. As a result, they are weaker than larger cities in attracting various innovation factors and resources, which inhibits Xuzhou's leading role in innovation to a certain extent.

## Further analysis

To further answer by what mechanism Xuzhou's innovation leadership effect can improve the innovation capacity of neighboring cities, according to value chain theory, we tested the effect of innovation capacity in three stages—(5) knowledge innovation primacy, (6) R&D innovation primacy, and (7) industrial innovation primacy—and analyzed the role of the innovation leading effect channels from the (8) industrialization level.

The leading role of Xuzhou in innovation is reflected in enhancing the leading position of surrounding cities in knowledge innovation, R&D innovation and industrial innovation and thus improving the innovation level of surrounding cities. In addition, enhancing the industrialization level of surrounding cities effectively promotes their innovation capacity.

The leading role of Xuzhou in innovation is reflected in enhancing the leading position of surrounding cities in knowledge innovation, R&D innovation and industrial innovation and thus improving the innovation level of surrounding cities. In addition, enhancing the industrialization level of surrounding cities effectively promotes their innovation capacity. Specifically (seen as Table 5), (i) knowledge innovation primacy refers to the degree to which a city leads in terms of knowledge output, transformation of scientific and technological achievements and talent attraction. Xuzhou city can help its neighboring cities improve their knowledge innovation capacity by establishing cooperative alliances with research institutions, universities and science and technology enterprises to carry out technological exchanges and talent cultivation; thus, Hypothesis 2 is supported. (ii) R&D innovation primacy refers to the degree to which a city is led in terms of scientific research investment, technological innovation and

**Table 5. Further results for innovation-led core cities in the Huaihai Economic Zone.**

| Variables | (5) | (6) | (7) | (8) |
|---|---|---|---|---|
| Innovled | 0.0374*** | 0.0202*** | 0.0263*** | 0.0452*** |
| | (0.0071) | (0.0040) | (0.0065) | (0.0124) |
| rpgdp | 13.1981*** | -1.7950 | -3.8009 | -2.8812 |
| | (4.2010) | (2.3764) | (3.8308) | (7.3532) |
| io | -13.6493* | 4.8973 | -13.0726** | -22.1539* |
| | (6.8516) | (3.8758) | (6.2479) | (11.9928) |
| fin | 1.5344 | -9.1140*** | -10.0375* | 47.7604*** |
| | (5.8296) | (3.2977) | (5.3160) | (10.2040) |
| tech | 1.5344 | -9.1140*** | -10.0375* | 47.7604*** |
| | (5.8296) | (3.2977) | (5.3160) | (10.2040) |
| Constant | 4.8847* | 2.5557*** | 7.3159 | 5.4172* |
| | (2.7784) | (0.6830) | (7.4986) | (2.7132) |
| i.city | control | control | control | control |
| i.year | control | control | control | control |
| N | 77 | 77 | 77 | 77 |
| $R^2$ | 0.8904 | 0.9489 | 0.9281 | 0.9468 |

product development. Xuzhou city can cooperate with enterprises in neighboring cities to carry out joint R&D projects and share technological resources to enhance the R&D innovation capacity of neighboring cities; thus, Hypothesis 3 is proved. (iii) Industrial innovation primacy refers to the degree to which a city leads in industrial structure adjustment, emerging industry cultivation and industrial upgrading. Xuzhou City can guide the enterprises in neighboring cities to carry out industrial upgrading and transformation through platforms such as industrial parks and science and technology parks to promote the improvement of the industrial innovation capacity of neighboring cities; thus, Hypothesis 4 is proved. (iv) The level of industrialization refers to the degree of industrialization and the level of industrial development of the city. Regions with high industrialization levels may be more likely to have advanced production technologies and management experience, thus promoting the enhancement of innovation capacity. In some cases, overindustrialization may lead to the overconcentration of resources and the inhibition of innovation capacity, resulting in a negative relationship. In the context of the Huaihai Economic Zone, the current level of industrialization has not yet reached a value inflection point. Xuzhou city can help its neighboring cities upgrade their industrialization level by introducing advanced production technologies and management experience, which will promote industrial upgrading and innovative development in the whole region and thus play a leading role in innovation. As a result, the mediating effect of the industrialization level is reflected, and hypothesis 5 is proved. In the future, Xuzhou city should increase in size, accelerate the improvement of strength, hub and radiation advantages, and make the advantages of being the first city more obvious and more influential.

## Conclusions and discussions

### Conclusions

Based on the theory of innovation chains, this paper comprehensively evaluates and analyses the innovation primacy of the Huaihai Economic Zone by referring to the evaluation index system of other scholars. In general, after becoming the core city of the Huaihai Economic Zone, Xuzhou's innovation primacy has been greatly improved, which has a certain siphon effect on the surrounding area and has a preliminary radiation-driving effect on the surrounding area. However, there are also several problems, such as a decrease in the market transformation of theoretical innovation and a weakness in industrial innovation and R&D innovation. The main conclusions and findings are as follows:

1. Ranking of innovation ability: According to the average score from 2010 to 2020, Xuzhou City ranks the highest in innovation ability, and its innovation ability score is the highest, followed by Jining, Suzhou, Huaibei and other cities. This shows that Xuzhou is leading in the field of innovation within the economic region.

2. Innovation leadership effect: Xuzhou City is recognized as an innovation leader in the whole region, and its innovation leadership effect remained strong in 2010, 2015, and 2020; after 2015, its innovation leadership effect rose sharply, indicating that Xuzhou city's strategic investment in science and technology innovation has achieved remarkable results.

3. City size impact: Research also shows that there is a positive correlation between city size and the innovation leadership effect and that larger cities are more likely to have a leading role in innovation.

4. Influence of control variables: In terms of control variables, the research finds that the level of technological development is positively correlated with the innovation leadership effect, while the degree of economic openness and the level of financial development are

negatively correlated with the innovation leadership effect. This finding suggests that cities exhibit stronger innovation leadership in terms of technology investment.

5. Xuzhou City plays a leading role in innovation among the surrounding cities, improving the leading position of knowledge innovation, R&D innovation and industrial innovation in the surrounding cities. Moreover, improving the industrialization level of surrounding cities also promotes the innovation capacity of these cities.

In brief, Xuzhou City, which is in the Huaihai economic region and is in the leading position in terms of innovation, has made remarkable progress, especially in terms of scientific and technological innovation. Moreover, city size and other factors also affect the innovation leadership effect. Xuzhou city can help its neighboring cities improve their knowledge, R&D and industrial innovation capacity by establishing cooperative alliances, sharing resources and introducing advanced technologies to achieve innovation-leading effects and promote the innovative development of the whole region. These conclusions provide an in-depth understanding of the innovation ability and leadership effect of cities in the Huaihai economic region and provide an important reference for future policymaking.

## Countermeasures

As the central city of the Huaihai economic region, Xuzhou should further enhance the primacy of innovation, enhance the leading force of innovation, and play a role in resource agglomeration in the whole region and in radiation to surrounding cities. The following strategies can be adopted. First, we can continuously create an innovative environment and atmosphere and enhance the primacy of R&D innovation [32]. We will implement an industrial technology research and development center with enterprises as the main body and universities as the auxiliary, encourage universities, enterprises and scientific research institutions to jointly carry out technology research and development and project innovation, and vigorously promote the integration of production, study, research and application and improve the effective number of invention patents, patent authorizations, patent applications and the efficiency of achievement transformation. In addition, we must rely on rich science and education resources to build an important national science and education cultural center, provide more high-quality talent for independent innovation and entrepreneurship development, and provide a strong guarantee for R&D innovation [12]. Second, industrial upgrading should be promoted, and industry primacy should be improved [33]. Xuzhou city should fully exploit its industrial advantages and characteristics. Xuzhou is an industrial center, so it is necessary to adhere to the advantages and characteristics of the first industry represented by Xuzhou Machinery. Second, Xuzhou city should fully integrate resources. The government should provide financial and policy support to guide resource integration, complementary advantages, and strong alliances to cultivate leading high-quality enterprises [12, 34]. In addition, we should strive to build a new national industrial base and industrial science and technology innovation center with global competitiveness, vigorously develop and cultivate hightech enterprises, and comprehensively promote the upgrading and transformation of the industrial structure and healthy economic development. Finally, we should seize development opportunities and actively create several high-tech innovative enterprises through the innovation and development of high-tech enterprises to drive and promote the development of regional hightech industries [3, 35].

The overall strength of the cities in the Huaihai Economic Zone is relatively balanced. The Huaihai Economic Zone focuses on the advantages of the central city Xuzhou, which is exploiting each city, has coordinated development, and has enhanced competitiveness and

influence on the Huaihai Economic Zone. This paper proposes the following suggestions. First, there is an urgent need to better integrate the strategic planning of the Huaihai Economic Zone and enhance the overall regional innovation capacity. Regional central cities need to be repositioned in combination with urban characteristics, make use of urban advantages, formulate and implement integrated strategic planning in the Huaihai Economic Zone, adjust industrial structure, scientifically integrate resources, realize complementary advantages for cities in the whole region, share development, and ultimately achieve win–win cooperation. Second, it is important to realize regional coordinated development, complementary advantages, and linkage development. The cities in the Huaihai Economic Zone should further enhance the balance and coordination of development; focus on the regional integration strategy of the Huaihai economy; closely link provinces and cities; strengthen policy coordination; plan and construct several basic, key and regional major projects; and promote the formation of a new situation of regional coordinated development. Moreover, by supporting policies, increasing policy implementation, promoting the formation of regional integration mechanisms, cooperating and sharing from industrial integration and infrastructure layout and other aspects, and developing regional linkages and integrations, more attention should be given to ultimately achieving regional coordination and common development. Third, promoting the sustainable development of the regional economy is the key problem. The Huaihai economic region should promote the upgrading of traditional industries and accelerate the development of the modern service industry. We should bravely to eliminate backward industries and promote transformation, actively explore new impetuses for high-quality economic development, and actively build a digital economic system to achieve balanced development of the population, economy, resources and environment. We should also promote higher quality development in the entire Huaihai economic region to achieve comprehensive and sustainable development.

## Deficiencies and prospects

Although some improvements have been made, due to data collection problems, there are still some blind spots that have not been examined, such as the introduction of talent. Factors such as the input of innovative resources have not been well reflected in the text. Second, due to the problem of its source, the data in this paper are not updated to include the latest 2022 data, and some of the latest policies of Xuzhou City are not included in the text. Third, this paper does not study all the cities in the overall region of the Huaihai economy; rather, only the cities in the core area of the Huaihai economic region are the research object, and the choice is somewhat narrow. The Huaihai economic zone is one of the earliest regional economic cooperation organizations in China. It consists of twenty prefecture-level cities in Jiangsu, Shandong, Henan and Anhui Provinces. This study selected eight cities in the core area of the Huaihai Economic Zone as representatives for research, both because of the availability of data and workload factors and because of the special status of the core area of the Huaihai Economic Zone. The geographical location of cities in the core area is important, but the region is located in the economic depression zone of the four provinces. Accelerating the development and revitalization of a region is very important for promoting the coordinated development of the regional economy. In view of the above limitations, follow-up studies will further expand the breadth of data collection and update the existing data around these issues.

## Acknowledgments

The authors are very grateful to the anonymous reviewers and editor for their insightful comments, which helped us to improve the quality of this paper.

## Author Contributions

**Data curation:** Qin-Xia Liu.

**Formal analysis:** Qin-Xia Liu.

**Funding acquisition:** Qin-Xia Liu.

**Investigation:** Qin-Xia Liu.

**Methodology:** Qin-Xia Liu.

**Writing – original draft:** Qin-Xia Liu.

**Writing – review & editing:** Qin-Xia Liu.

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
