## [Decision Letter · Decision Letter 0]

12 Dec 2023

PONE-D-23-33941Evaluation on the Innovation Primacy in Cross-regional Center Cities: Evidence from Huaihai Economic Zone in China (2010-2020)PLOS ONE

Dear Dr. Liu,

Thank you for submitting your manuscript to PLOS ONE. After careful consideration, we feel that it has merit but does not fully meet PLOS ONE’s publication criteria as it currently stands. Therefore, we invite you to submit a revised version of the manuscript that addresses the points raised during the review process.

We look forward to receiving your revised manuscript.

Kind regards,

Aamna Mohammed AlShehhi, PhD

Academic Editor

PLOS ONE

Journal Requirements:

"General Project Subjects of Philosophy and Social Science Research in Universities of Jiangsu Province"

"The authors are very grateful to the anonymous reviewers and editor for their insightful comments that helped us 

sufficiently improve the quality of this paper. This research is supported by General Project Subjects of Philosophy and 

Social Science Research in Universities of Jiangsu Province, Evaluation and influencing factors of green technology 

innovation of listed enterprises in Jiangsu Province under the background of digital transformation (Grant No.

2023SJYB1177)."

"General Project Subjects of Philosophy and Social Science Research in Universities of Jiangsu Province"

Reviewers' comments:

Reviewer's Responses to Questions

**Comments to the Author**

1. Is the manuscript technically sound, and do the data support the conclusions?

Reviewer #1: Yes

Reviewer #2: Yes

Reviewer #3: Yes

2. Has the statistical analysis been performed appropriately and rigorously? 

Reviewer #1: Yes

Reviewer #2: Yes

Reviewer #3: I Don't Know

3. Have the authors made all data underlying the findings in their manuscript fully available?

Reviewer #1: Yes

Reviewer #2: Yes

Reviewer #3: Yes

4. Is the manuscript presented in an intelligible fashion and written in standard English?

Reviewer #1: Yes

Reviewer #2: Yes

Reviewer #3: No

5. Review Comments to the Author

Reviewer #1: Review Report for Manuscript ID: PONE-D-23-33941. “Evaluation on the Innovation Primacy in Cross-regional Center Cities: Evidence from Huaihai Economic Zone in China (2010-2020)” I applaud the authors for their efforts in crafting this article. However, there are a number of issues with this paper that need to be revised. These concerns are delineated as follows:

1 In the first line of page 2, the author omitted a little information. Like, “In recent years, scholars such as [9,10]have gradually focused on the primacy of innovation.”. There are many other similar mistakes in this article.

2 Could you provide a detailed comparative analysis of your study's empirical results in relation to findings from existing literature. Highlight the distinctive aspects or divergences in your results compared to other studies, offering an in-depth empirical discussion to contextualize and interpret these differences.

3 Please give a mechanistic analysis of this conclusion. “Xuzhou can promote the improvement of the innovation capability of surrounding cities through the innovation leadership effect.”

My Recommendation: Minor Revision

Reviewer #2: 1. The description of the background of the article should be strengthened to emphasize why the topic should be studied.

2. In order to ensure the robustness of empirical results, this paper should consider endogeneity and other issues.

3. This paper should analyze the limitations of this paper and put forward prospects at the end of the paper.

Reviewer #3: The author should pay more attention to the expression. It will be better with polish. The References list is not consistent in format. Some missing information when retrieved from the database needs to be curated. Most of the work is declarative, lacking author's comments. Open questions should be proposed for future research. Send to a more oriented journal would be more suitable.

With thank you.

6. PLOS authors have the option to publish the peer review history of their article (what does this mean?). If published, this will include your full peer review and any attached files.

Reviewer #1: No

Reviewer #2: No

Reviewer #3: No

---

## [Author Response · Author response to Decision Letter 0]

15 Jan 2024

Thanks a lot for your kind and helpful suggestion!

---

## [Decision Letter · Decision Letter 1]

21 Feb 2024

Evaluation of innovation primacy in cross-regional central cities: Evidence from the Huaihai Economic Zone in China (2010-2020)

PONE-D-23-33941R1

Dear Dr. Qin-Xia Liu,

We’re pleased to inform you that your manuscript has been judged scientifically suitable for publication and will be formally accepted for publication once it meets all outstanding technical requirements.

Kind regards,

Aamna Mohammed AlShehhi, PhD

Academic Editor

PLOS ONE

Additional Editor Comments (optional):

Reviewers' comments:

Reviewer's Responses to Questions

**Comments to the Author**

1. If the authors have adequately addressed your comments raised in a previous round of review and you feel that this manuscript is now acceptable for publication, you may indicate that here to bypass the “Comments to the Author” section, enter your conflict of interest statement in the “Confidential to Editor” section, and submit your "Accept" recommendation.

Reviewer #1: All comments have been addressed

Reviewer #2: (No Response)

2. Is the manuscript technically sound, and do the data support the conclusions?

Reviewer #1: Yes

Reviewer #2: Yes

3. Has the statistical analysis been performed appropriately and rigorously? 

Reviewer #1: Yes

Reviewer #2: Yes

4. Have the authors made all data underlying the findings in their manuscript fully available?

Reviewer #1: Yes

Reviewer #2: Yes

5. Is the manuscript presented in an intelligible fashion and written in standard English?

Reviewer #1: Yes

Reviewer #2: Yes

6. Review Comments to the Author

Reviewer #1: (No Response)

Reviewer #2: (No Response)

7. PLOS authors have the option to publish the peer review history of their article (what does this mean?). If published, this will include your full peer review and any attached files.

Reviewer #1: No

Reviewer #2: No

---

## [Editor Report · Acceptance letter]

24 Feb 2024

PONE-D-23-33941R1 

PLOS ONE

Dear Dr. Liu, 

I'm pleased to inform you that your manuscript has been deemed suitable for publication in PLOS ONE. Congratulations! Your manuscript is now being handed over to our production team.

Kind regards, 

on behalf of

Dr. Aamna Mohammed AlShehhi 

Academic Editor

PLOS ONE